# OpenReview forum: "EEG-Language Pretraining for Highly Label-Efficient Pathology Detection"
_ICLR.cc/2025/Conference — Submitted to ICLR 2025_

### Official Review · Reviewer_8ZnL · 2024-11-02

**Soundness:** 3
**Presentation:** 3
**Contribution:** 2
**Rating:** 5
**Confidence:** 4

**Summary:**

The paper presents an EEG pretraining LLM for disease detection, introducing a novel method that integrates multimodal alignment with EEG time series cropping and text segmentation. The experimental results demonstrate the superior performance of the proposed method compared to existing approaches. Additionally, the paper is well organized, and the references to recent works are comprehensive.

**Strengths:**

1.	The aligning of EEG crops and the text segments is quiet good idea.
2.	The proposed method seems to achieve superior performance of the proposed method compared to existing approaches.
3.	This paper is well organized and with clear presentation.

**Weaknesses:**

1.	Classifying EEG signals into [normal] or [abnormal] is relatively straightforward, even with traditional (non-large) models. Using such high-resource cost model to do such a simple work is nor enough.
2.	The authors rely on a pretrained language model for the text encoder without fine-tuning it on specialized medical data. This approach may hinder performance in the medical domain, as the pretrained model likely lacks the necessary medical knowledge and context.
3.	While the authors employ a Multiple Instance Learning (MIL) approach to align EEG segments with text, the generality of the provided text segments (e.g., "clinical history," "medications") raises questions. Specific text segments may not be accurately aligned with corresponding EEG segments, potentially undermining the effectiveness of the alignment strategy.

**Questions:**

1. The authors state, “As paired medical EEG data and clinical reports are scarce, training the text encoder function fl from scratch is unlikely to be successful… To prevent resulting information loss, we follow the recommendations by Liu et al. (2023a) to use a pretrained language model for fl and freeze its weights during training.” However, I believe that the scarcity of medical reports actually suggests the need to fine-tune the text encoder. The data used to pretrain large language models likely consists of general dialogue, lacking specialized medical knowledge and explanations. Without fine-tuning on medical data, the model’s performance in this domain may be suboptimal.
2. The use of MIL to align EEG segments with corresponding text segments is an interesting approach. The paper states: “While previous approaches aim to align text and EEG crops uniformly, certain text segments likely describe specific EEG sections more accurately than others.” Can I interpret this with the following example:
•	Clinical Text: “One segment of the EEG data indicates deep sleep, while another shows the patient gradually waking up.”
•	EEG: One crop represents deep sleep; the other represents waking up.
Therefore, are the authors aligning the first EEG crop with the “deep sleep” text and the second with the “waking up” text? If so, why are the text segments provided in the Figure 1 more general, such as "clinical history," "medications," and "description of EEG," which seem to describe the EEG comprehensively rather than specific segments?
3. Classifying EEG signals into [normal] or [abnormal] is relatively straightforward, even with traditional (non-large) models. Are the results in Table 3 quite low? Have the authors compared their performance with other large models or traditional models?
4. Given the high cost of training such a large model, I believe it’s important to set higher expectations for its capabilities. Have the authors considered multi-disease recognition or generating other types of medical records?

---

> ### Author Response · Authors · 2024-11-19
> **Reply to Reviewer 8ZnL (1 out of 2)**
>
> We sincerely thank the reviewer for their comments and questions. We are pleased to hear our novel extensions to multimodal modeling are appreciated. We would like to respond to the raised weaknesses and questions in order. For your convenience, in addition to the revision we have collected the new tables and figures in [this anonymous document [link]](https://docs.google.com/document/d/e/2PACX-1vSsGqVFkvY1PEKzMxvVhHX-UiJ_r_1zS1dbF0RBf_Xlgj6D7QVwkncYkj2MXYR98sNT6bVA9Po9kpBl/pub).
>
>
> > **W1**: Classifying EEG signals into [normal] or [abnormal] is relatively straightforward, even with traditional (non-large) models. Using such high-resource cost model to do such a simple work is nor enough.
>
> We appreciate the importance of cost-benefit trade-offs and would like to clarify two points. First, as we use pretrained language models and only optimize an MLP text projector, the training is actually faster than EEG-only pretraining with augmentations which add significant computational overhead. Once a model is trained (under 7 hours on a single RTX 3090, which is 4 year old hardware), inference with a model is fast and requires little hardware. We believe that this, in combination with the considerable performance gains, make these approaches viable for future clinical applications. The normal/abnormal distinction constitutes an important aspect of hospital EEG evaluations.
>
> Second, we are pleased to share additional model evaluations on more challenging tasks below.
>
> **1. External validation using the NMT Dataset** - Section A.1 (Page 15)
>
> We use the NMT Scalp EEG Dataset from Pakistan as the second largest EEG corpus with abnormality labels to evaluate models out-of-distribution. This constitutes a challenging evaluation as the dataset is highly imbalanced and differs considerably from TUEG in terms of demographics and EEG recording setup. We find that our ELM-MIL approach still performs best in pathology detection across most settings.
>
> **2. Additional baseline task: Seizure detection using TUSZ** - Section A.2 (Page 15)
>
> We evaluate models on the more challenging task of seizure detection based on a single, short EEG crop by pretraining all methods on 5-second crops. We observe that ELM-MIL exhibits performance improvements over EEG-only pretraining also in this considerably different evaluation.
>
> **3. Additional baseline task: 6-class event detection using TUEV** - Section A.2 (Page 15+)
>
> TUEV poses a classification problem with three types of pathology events as well as eye movements, artefacts, and background activity, based on single 5-second EEG crops. We observe good performance of ELM-MIL on this task too.
>
> We additionally hope that the reviewer finds the additional class-specific visualizations for TUEV insightful (Figure 4; Page 17).
>
> > **W2**: The authors rely on a pretrained language model for the text encoder without fine-tuning it on specialized medical data. This approach may hinder performance in the medical domain, as the pretrained model likely lacks the necessary medical knowledge and context.
>
> We thank the reviewer for raising this point and we would like to clarify that all three language models compared in section B.3 (Table 10) are based on models trained from scratch on medical data (PubMed and/or electronic health records). Nevertheless, it would certainly be interesting for future work to investigate whether sufficient EEG-specific text can be gathered for further finetuning. However, this would also significantly raise the computational costs of the approach.
>
> > **W3**: On the generality of text segments
>
> We appreciate this concern, which is also why we decided to include extensive analyses on how to perform the text segmentation. We were pleased to find that while omitting important pathology information hurts performance, the method proves robust against including lower signal-to-noise segments. Nevertheless, follow-up work could indeed investigate additional strategies for dealing with the heterogeneous medical reports.
>
> We will respond to the further points in a second reply.

---

> ### Author Response · Authors · 2024-11-19
> **Reply to Reviewer 8ZnL (2 out of 2)**
>
> > **Q1**:  (...) I believe that the scarcity of medical reports actually suggests the need to fine-tune the text encoder. The data used to pretrain large language models likely consists of general dialogue, lacking specialized medical knowledge and explanations. Without fine-tuning on medical data, the model’s performance in this domain may be suboptimal.
>
> We thank the reviewer for their question. We comment on the pretraining of the language models themselves above, for which only medical data was used. In terms of finetuning language models during multimodal training, we agree that this is a worthwhile research question.  Afterall, our results indicate that the inclusion of a text projector is helpful and it is possible finetuning of the text encoder itself could be too. However, we expected full finetuning to only provide marginal improvements due to existing empirical evidence such as Liu et al. as well as the risk of overfitting on our limited dataset. This, in combination with the fact that it increases the resource-costs for the ELMs immensely and our constrained available computational resources, we decided to pursue the other research questions in this initial work first.
>
> > **Q2**: (...) why are the text segments provided in the Figure 1 more general?
>
> We appreciate the opportunity to elaborate on this important observation. One of the challenges of EEG-language modeling indeed relates to the fact that some parts of the medical reports describe very general factors (e.g. demographic information or global clinical EEG features such as persistent slowing) while also describing local events (e.g. seizures, spike and sharp waves). The data sharing contracts prevent us from printing possible report information, which is why we refer to (clusters of) headings. We try to clarify this in Figure 13 (Section E.3 Page 25). For example, "Description of the Record" includes headings such as "Events" or "Epileptiform Activity" describing local effects, whereas a "Clinical Correlation" heading from the "Interpretation" may describe broader characteristics. Our MIL approach aims to allow flexibility in terms of aligning EEG crops more uniformly (global information) or sparsely (local information).
>
> > **Q3**: Classifying EEG signals into [normal] or [abnormal] is relatively straightforward, even with traditional (non-large) models. Are the results in Table 3 quite low? Have the authors compared their performance with other large models or traditional models?
>
> We thank the reviewer for their question. As Table 3 shows zero-shot classification, we would like to note that to our knowledge we are the first to show such results without needing any labelled data, which other deep models or traditional models are not capable of. For ELM-MIL e,l, this is a somewhat stronger result than for example in the original computer vision CLIP paper by Radford et al. (2021), where only n=4 per class is needed for the linear probe to match zero-shot classification while here n>14 is needed (as linear probing on 1% TUAB is below zero-shot for ELM-MIL e,l).
>
> For many configurations, it is true that zero-shot performance is poor, highlighting the importance of choosing which text should be aligned, which we analyse extensively. Nevertheless, given bidirectional alignment with all text clusters (ELM-MIL e,l), alignment is stable across seeds and temperature parameter values.
>
> As zero-shot performance has not been shown before, we could compare the performance of 84.31% accuracy to 1% linear probing by Mohsenvand et al. (2021), which find 78.53% and 77.31% performance of their large models trained across multiple datasets. Please let us know if you would like to see further comparisons.
>
> > **Q4**: Given the high cost of training such a large model, I believe it’s important to set higher expectations for its capabilities. Have the authors considered multi-disease recognition or generating other types of medical records?
>
> We agree and believe that generating medical reports based on EEG data is a fascinating avenue, which our current work suggests may be possible.  We leave this for follow-up work but we are pleased to be able to provide the additional evaluations on NMT, TUSZ, and TUEV showing broader model capabilities. We hope to have addressed the concern of high model training costs in the comment above, as we have avoided high costs by freezing weights of language models which are specifically pretrained on medical data.

---

### Official Review · Reviewer_37ck · 2024-11-02

**Soundness:** 3
**Presentation:** 3
**Contribution:** 3
**Rating:** 8
**Confidence:** 4

**Summary:**

The paper presents EEG-Language Models (ELMs) pretrained jointly on clinical reports and EEG data using a multiple instance learning (MIL) variant of contrastive learning (MIL-NCE). The authors evaluate these models on downstream tasks, including retrieval analysis and pathology classification. Pathology classification is performed using both zero-shot and fine-tuning approaches. The model’s performance is compared against EEG-only pretraining methods, which serve as the main baseline, and demonstrates significant improvements across all evaluations. The paper also explores various ELM configurations and finds that their ELM-MIL variant achieves the best overall performance.

**Strengths:**

- While multiple instance learning and contrastive learning are not novel in isolation, the paper adapts these techniques to the novel domain of brain activity signals and demonstrates substantial benefits over using EEG-only pretraining methods. The results show that exposing the model to a variety of report segments enables the learning of richer representations.

- The paper includes thorough ablation studies that explore various design choices, such as the integration of multiple instance learning and different segments of clinical text. The model is also compared against various comparable baseline approaches, including BYOL and ContraWR, with standard deviations reported over multiple runs to indicate the confidence level of their results.

- Overall, the paper is well-written and provides robust baseline comparisons, detailed ablations of design choices, and a strong justification for incorporating text into the pretraining of EEG foundation models.

**Weaknesses:**

- While the paper focuses on pretraining strategies, the results would be more robust if a supervised baseline were included in Table 2. For example, training a neural network to classify pathology directly from raw EEG signal space could provide a useful comparison. Ideally, this supervised model should use the same architecture as the EEG encoder employed by the authors.

- The impact of the temperature parameter on performance, as shown in Figure 3, is not entirely clear. Including performance results for more temperature parameter values could provide better insight into its effect on the model.

- The tasks used to evaluate the models are relatively narrow and not very challenging. Although the paper demonstrates performance improvements over the baseline on these benchmarks, it is difficult to conclude the method’s overall efficacy without evaluating on more complex tasks. However, it is important to recognize the difficulty in obtaining EEG-report datasets. One suggestion is to explore sleep databases for potential paired EEG and report data, which might include sleep apnea or sleep stage annotations. These annotations could be converted into brief report summaries based on the severity of apnea, for instance.

**Questions:**

- Why is there no supervised baseline for pathology classification?

- The temperature ablation (Figure 3) is somewhat unclear. Including results for additional temperature values could enhance the clarity of the plot.

- Where do the authors see this approach being most useful? What other types of tasks could benefit from this method compared to baseline approaches?

---

> ### Author Response · Authors · 2024-11-19
> **Reply to Reviewer 37ck**
>
> We sincerely thank the reviewer for their comments and questions. We are pleased that our novel approach to EEG pretraining, thorough ablation studies, and robust comparisons are appreciated. We would like to respond to the raised weaknesses and questions in order. For your convenience, in addition to the revision we have collected the new tables and figures in [this anonymous document [link]](https://docs.google.com/document/d/e/2PACX-1vSsGqVFkvY1PEKzMxvVhHX-UiJ_r_1zS1dbF0RBf_Xlgj6D7QVwkncYkj2MXYR98sNT6bVA9Po9kpBl/pub).
>
>
> > **W1**: Supervised baselines
>
> We thank the reviewer for the suggestion and now include a supervised baseline for which we use the same EEG encoder architecture. For TUAB (Table 2; Page 8) as well as the additional new evaluations, we note that ELMs perform well compared to this baseline.
>
> > **W2**:  Including performance results for more temperature parameter values could provide better insight into its effect on the model.
>
> We appreciate this suggestion and we have extended these analyses to include a wider range of parameter values. We believe it now more clearly highlights the greater stability of our MIL variant, particularly for zero-shot classification (Figure 3; Page 10).
>
> > **W3**: Additional benchmarks
>
> We thank the reviewer for highlighting this issue. We agree that sleep databases constitute a highly interesting extension. However, we decided to stay closer to the pathology scope of the paper by evaluating models out-of-distribution on the NMT dataset as well as perform more complex pathology tasks. We hope these nevertheless address the reviewer's wishes.
>
> **1. External validation using the NMT Dataset - Section A.1 (Page 15)**
> We use the NMT Scalp EEG Dataset from Pakistan as the second largest EEG corpus with abnormality labels to evaluate models out-of-distribution. This constitutes a challenging evaluation as the dataset is highly imbalanced and differs considerably from TUEG in terms of demographics and EEG recording setup. We find that our ELM-MIL approach still performs best in pathology detection across most settings.
>
> **2. Additional evaluation: Seizure detection using TUSZ - Section A.2 (Page 15)**
> We evaluate models on the more challenging task of seizure detection based on a single, short EEG crop by pretraining all methods on 5-second crops. We observe that ELM-MIL exhibits performance improvements over EEG-only pretraining also in this considerably different evaluation.
>
> **3. Additional evaluation: 6-class event detection using TUEV - Section A.2 (Page 15+)**
> TUEV poses a classification problem with three types of pathology events as well as eye movements, artefacts, and background activity, based on single 5-second EEG crops. We observe good performance of ELM-MIL on this task too.
>
> We additionally hope that the reviewer finds the additional class-specific plots for TUEV insightful (Figure 4; Page 17).
>
> > **Q1, Q2**
>
> We hope to have sufficiently addressed questions 1 and 2 in the discussion of the weaknesses.
>
> > **Q3**: Where do the authors see this approach being most useful? What other types of tasks could benefit from this method compared to baseline approaches?
>
> We thank the reviewer for this interesting question. We hope for several high-impact applications for this approach. The method is particularly valuable for EEG analysis, where it can support both pathology detection and event monitoring while leveraging the rich temporal information in clinical reports. While our work focused on EEG, the approach could be extended to other neuroimaging modalities where clinical descriptions are routinely collected. For instance, structural MRI, which is a common screening tool, could benefit from learning representations that bridge imaging features with radiological reports.
>
> The method could be especially useful for longitudinal data analysis. While fMRI is increasingly used in pre-surgical planning, it will still be more constrained in terms of available training data. In contrast, our approach could be particularly valuable for continuous monitoring scenarios, such as analyzing long-term EEG recordings or data from implanted devices (e.g., Deep Brain Stimulation systems), where it could help align neural activity patterns with clinical observations and patient history from electronic health records.
> The key advantage over baseline approaches is the ability to learn from unstructured clinical descriptions, potentially capturing subtle patterns and relationships that might be missed in traditional supervised learning approaches that rely on discrete labels.

---

> > ### Comment · Reviewer_37ck · 2024-11-24
> >
> > Thank you for addressing my concerns. I do not have any further questions. I have adjusted the score accordingly.

---

### Official Review · Reviewer_KPnm · 2024-11-03

**Soundness:** 3
**Presentation:** 3
**Contribution:** 3
**Rating:** 6
**Confidence:** 4

**Summary:**

This article explores a multimodal model that integrates natural language with EEG brain data, introducing EEG-Language Models (ELMs) trained on clinical reports and EEG data. By using time-series cropping and text segmentation, the model better aligns EEG and text data, showing significant advantages in pathology detection, especially when data is limited. The study demonstrates that this combined approach improves detection accuracy and presents new possibilities for medical applications.

**Strengths:**

1. Richer Representation Learning: Through time-series cropping and text segmentation, this method increases the number of samples, allowing the model to learn richer and more diverse representations. This improved data variety helps the model capture complex patterns in EEG and clinical text, enhancing performance in pathology detection tasks and making it more adaptable to different clinical scenarios.

2. Data Preprocessing and Noise Filtering: The method employs careful data preprocessing and cropping to remove irrelevant noise, focusing on the most important EEG signals. This improves the efficiency of training and inference by ensuring the model works with high-quality, meaningful data, leading to more accurate and reliable performance in clinical settings.

**Weaknesses:**

1. Limited Dataset Scale: Despite the demonstrated potential of multimodal pretraining in medical EEG analysis, the model remains significantly constrained by the limited availability of clinical EEG data and corresponding textual reports. The scarcity of such data hinders the model's generalization capability, particularly when dealing with unseen pathological cases, where performance may degrade substantially. Additionally, the small dataset size increases the risk of overfitting to specific pathological patterns, reducing the model’s robustness and applicability across diverse clinical settings. This limitation also raises concerns about the model’s transferability and consistency when deployed in different hospitals or patient populations.

2. Potential Bias Risks: The method's reliance on pretrained language models and text summarization introduces the possibility of inherited biases or inaccuracies from the original language models, which are often trained on general, non-medical data. These biases may be exacerbated or propagated in the EEG representations and subsequent clinical decision-making. For instance, if the pretrained model exhibits particular biases toward certain symptoms or pathologies, it could result in unbalanced predictions during EEG analysis. Moreover, the text summarization process might oversimplify or misrepresent critical information, further amplifying bias effects. Combined with the framework’s lack of innovation—relying on basic contrastive learning mechanisms—this approach may be insufficiently adaptable for more complex or dynamic clinical scenarios.

3. Alignment Instability: Although the implementation of a multiple-instance learning strategy helps mitigate some of the alignment issues between EEG data and text segments, it does not fully resolve the challenges posed by mismatched or weakly correlated data. In practice, clinical reports and EEG signals may contain partially irrelevant or loosely related information. For instance, certain report sections might include background details unrelated to the EEG signals, or critical EEG features may not be explicitly referenced in the text. Such discrepancies can lead to unstable multimodal alignment, reducing the model’s accuracy and robustness in pathology detection tasks. Consequently, the model's reliability and practical utility in real-world clinical environments may be compromised due to suboptimal alignment performance.

**Questions:**

1. Considering the inherent differences between EEG and fMRI, could you elaborate on the decision to use EEG data for this study? fMRI provides high spatial resolution and detailed information about brain regions involved in specific tasks or conditions, which could offer a more comprehensive and fine-grained understanding when aligned with clinical text. How might the use of fMRI improve the quality of alignment or representation learning in comparison to EEG, and what trade-offs would this involve in terms of data availability, preprocessing complexity, and computational demands?
2. EEG data is known for being highly susceptible to various forms of noise and artifacts, such as muscle activity, eye movements, and environmental interference. Effective preprocessing is crucial for ensuring the quality and reliability of the data used in model training. Could you specify the techniques employed for noise reduction and artifact removal? Additionally, what measures were taken to standardize and segment the EEG signals to make them suitable for training, and how were these techniques validated to ensure they do not compromise important neural information?
3. The current analysis mainly features cross-sectional comparisons between different methods. Would it be possible to add more thorough comparative experiments, perhaps including longitudinal analyses or additional baselines, to provide a more complete understanding of the model's performance and robustness?

---

> ### Author Response · Authors · 2024-11-19
> **Reply to Reviewer KPnm (1 out of 2)**
>
> We thank the reviewer for their elaborate review. We are pleased to hear our novel extensions to multimodal modeling are appreciated. We would like to respond to the raised weaknesses and questions in order. For your convenience, in addition to the revision we have collected the new tables and figures in [this anonymous document [link]](https://docs.google.com/document/d/e/2PACX-1vSsGqVFkvY1PEKzMxvVhHX-UiJ_r_1zS1dbF0RBf_Xlgj6D7QVwkncYkj2MXYR98sNT6bVA9Po9kpBl/pub).
>
>
> > **W1: Limited Dataset Scale**
>
> We appreciate the reviewer's concerns. We are pleased to provide external validation results using the NMT Scalp EEG Dataset, featuring a different EEG setup and demographics from a South Asian population (please see Section A.1 - Page 15). Specifically, the data was recorded from a South Asian population at the Pak-Emirates Military Hospital, Rawalpindi, Pakistan, using a different EEG recording setup. Furthermore, the NMT participants are considerably younger, feature more males (66.6\%), and their EEG recordings are labeled predominantly normal (83.8% in the training set). This enables a challenging and imbalanced external validation for representation learning methods. We observe that our ELM-MIL approach scores highest in all-but-one setting, indicating significant transferability of the learned representations.
>
> Furthermore, we would like to note that large amounts of this type of data exists in hospitals globally. Until now, there was little incentive to undertake the significant effort to share this data. We hope that promising research such as presented here may spur further data releases as method development progresses. Without these initial research efforts, it is unlikely for the status quo to change.
>
> > **W2: Potential Bias Risks: The method's reliance on pretrained language models and text summarization introduces the possibility of inherited biases or inaccuracies from the original language models, which are often trained on general, non-medical data. (...)**
>
> We would like to clarify that all three language models compared in section B.3 (Table 10) are models trained from scratch on medical data (PubMed and/or electronic health records). Indeed, these may still contain certain biases which could affect downstream performance, which should be investigated prior to clinical applications, together with potential biases present in EEG encoder with respect to demographics or pathologies.
>
> We thank the reviewer for the comment on the risk of text summarization. We note however that this was only one of multiple report processing strategies we compared and one which we did not use for any follow-up analyses. Instead, we conclude that using a variety of the original textual information is beneficial.
>
> Regarding the framework, we would like to take the opportunity to iterate that to our knowledge, no previous work has proposed to combine sub-unit alignment with a MIL strategy. As such, the methods of the present work enable the first application of multimodal language modelling to clinical brain signals. Moreover, we are pleased to provide new evaluations on the TUSZ and TUEV datasets (we expand on this below) which require clinical event detection based on 5-second EEG segments, showcasing that the methods are adaptable to deal with both recording-level summarization and event detection.
>
> > **W3: Alignment Instability**
>
> We appreciate that the medical reports also include less relevant data. This is why we decided to do extensive analyses on how to segment the clinical reports. Consequently, we find that models do not suffer on evaluations by expanding the text input and that rather omitting important sections is harmful. While we agree that the alignment process is noisy, if one aims to use a model to assist decision making in the future, representations learned by ELMs may nevertheless constitute the best available option for the initialization. Namely, the comparison is with respect to EEG-only pretraining, which we find ELMs tend to outperform.
>
> > **Q1: EEG vs fMRI**
>
>  We thank the reviewer for their question. Whereas many hospitals perform EEG assessments, we are not aware of such common practice to exist for fMRI, which is rarer and mostly limited for pre-surgical planning. The costs of operating the systems differ considerably, making EEG more widely available. Thus, currently EEG is the only option for investigating the utility of integrating text with functional brain data using large datasets. Finally, fMRI is generally associated with considerably more elaborate and complex preprocessing requirements as well as greater computational demands due to the significantly higher memory footprint of fMRI data.
>
> We will respond to the final points in in reply 2.

---

> ### Author Response · Authors · 2024-11-19
> **Reply to Reviewer KPnm (2 out of 2)**
>
> > **Q2**: EEG Artefacts
>
> The standard in the field of using deep learning for EEG has been to apply relatively light preprocessing. A significant part of the artefacts are fortunately straightforward to resolve. For example, regarding environmental interference, powerline noise can be reliably filtered out, and as it is widely agreed that a majority of signal in EEG is in the lower frequencies, this largely avoids compromising neural information. With respect to muscle activity and eye movements, these indeed often elicit significantly larger responses than neural activity. We use an amplitude-based cut-off which is almost certain not to erroneously reject neural information. We crop EEG segments into fixed-length windows and perform Z-score normalisation of data to aid training. We would like to stress that all these procedures are standard in neuroscience and clinical machine learning research.
>
> > **Q3**: Would it be possible to add more thorough comparative experiments, perhaps including longitudinal analyses or additional baselines, to provide a more complete understanding of the model's performance and robustness?
>
> We thank the reviewer for this suggestion and are pleased to provide three additional evaluations:
>
> **1. External validation using the NMT Dataset - Section A.1 (Page 15)**
> We use the NMT Scalp EEG Dataset from Pakistan as the second largest EEG corpus with abnormality labels to evaluate models out-of-distribution. This constitutes a challenging evaluation as the dataset is highly imbalanced and differs considerably from TUEG in terms of demographics and EEG recording setup. We find that our ELM-MIL approach still performs best in pathology detection across most settings.
>
> **2. Additional evaluation: Seizure detection using TUSZ - Section A.2 (Page 15)**
> We evaluate models on the more challenging task of seizure detection based on a single, short EEG crop by pretraining all methods on 5-second crops. We observe that ELM-MIL exhibits performance improvements over EEG-only pretraining also in this considerably different evaluation.
>
> **3. Additional evaluation: 6-class event detection using TUEV - Section A.2 (Page 15+)**
> TUEV poses a classification problem with three types of pathology events as well as eye movements, artefacts, and background activity, based on single 5-second EEG crops. We observe good performance of ELM-MIL on this task too.
>
> In regards to understanding the model's performance and robustness, we kindly refer the reviewer to the new result visualizations for single-class performance comparisons on TUEV (Section A.2 Page 15-17 and Figure 4 (Page 17)).

---

> > ### Comment · Reviewer_KPnm · 2024-11-19
> > **Response**
> >
> > Thank you for providing a detailed explanation in the rebuttal to address the questions I raised. Your response has significantly clarified my concerns regarding certain aspects of the paper. I believe these additional details enhance the clarity and contributions of the work, and I will adjust my score accordingly.

---

### Official Review · Reviewer_wPm6 · 2024-11-04

**Soundness:** 2
**Presentation:** 3
**Contribution:** 2
**Rating:** 5
**Confidence:** 4

**Summary:**

This paper proposes a method for training EEG-language model for pathology detection with EEG data. It leverages the success of multimodal language modeling in other domains, aiming to learn richer representations by aligning EEG time series with reports. Furthermore, the paper introduces sub-unit alignment using time series cropping and text segmentation to address the length discrepancy between modalities. It also propose a MIL extension to EEG and text data to handle inconsistencies in relevance and explore different text segmentation strategies based on report content.

**Strengths:**

- Applying multimodal language modeling to EEG and clinical text reports is a promising and much needed approach for zero-shot evaluation.
- Addresses the challenges posed by the length and heterogeneity of EEG and clinical text data through sub-unit alignment and MIL.
- Strong empirical results, especially in low-label regimes.
- Paper shows comprehensive ablation studies and analysis of different components.

**Weaknesses:**

- Limited evaluation on only one corpus (TUEG/TUAB). It would be interesting to see how the model generalizes/transfer to other datasets collected in a different setup/environment.
- Pretraining dataset with 15144 files are also relatively small compared to other domains. However, this issue is not related to the paper but more to lack of availability of high-quality large-scale EEG-Text data.
- The model performance seems highly sensitive to text segmentation and the type of text input (e.g., LLM summaries). This sensitivity raises concerns about the robustness of the approach and its applicability to other clinical settings.
- The comparison to EEG-only SSL methods is valuable, comparing the proposed model to fully supervised methods (when sufficient labeled data is available) is crucial for assessing the practical utility of the approach. I also see lack of comparison with other EEG-only self-supervised methods.

**Questions:**

- While the results show improved performance, the paper lacks a deeper discussion of the clinical implications of the learned representations. What kind of insights do ELMs offer for understanding EEG pathologies?
- I am wondering how does the proposed sub-unit alignment affect the temporal dependencies in EEG data? does it miss important segments relating to some specific conditions?
- Regarding the normalization by number of crops in MIL loss. Is this appropriately addressing the varying amount of information in different-length crops and text segments?
- I also have reservations over the procedure of selecting optimal value for temperature parameter. Why a separate held-out validation data is not used?

---

> ### Author Response · Authors · 2024-11-19
> **Reply to Reviewer wPm6 (1 out of 2)**
>
> We sincerely thank the reviewer for their valuable comments. We are pleased to hear that our multimodal approach is deemed promising with strong results. We would like to respond to the raised weaknesses and questions in order. For your convenience,  we collected the new tables and figures in [here [link]](https://docs.google.com/document/d/e/2PACX-1vSsGqVFkvY1PEKzMxvVhHX-UiJ_r_1zS1dbF0RBf_Xlgj6D7QVwkncYkj2MXYR98sNT6bVA9Po9kpBl/pub).
>
>
> > **W1: Limited evaluations**
>
> We appreciate the suggestion to evaluate out-of-distribution performance. We are pleased to provide external validation results, for which we use the NMT Scalp EEG Dataset from Pakistan as the second largest EEG corpus with abnormality labels to evaluate models out-of-distribution. This constitutes a challenging evaluation as the dataset is highly imbalanced and differs considerably from TUEG in terms of demographics and EEG recording setup. We find that our ELM-MIL approach still performs best in pathology detection across most settings.  Section A.1 (Page 15)
>
> We additionally provide two further evaluations:
>
> **Seizure detection using TUSZ - Section A.2 (Page 15)**
> We evaluate models on the more challenging task of seizure detection based on a single, short EEG crop by pretraining all methods on 5-second crops. We observe that ELM-MIL exhibits performance improvements over EEG-only pretraining also in this considerably different evaluation.
>
> **6-class event detection using TUEV - Section A.2 (Page 15+)**
> TUEV poses a classification problem with three types of pathology events as well as eye movements, artefacts, and background activity, based on single 5-second EEG crops. We observe good performance of ELM-MIL on this task too.
>
> > **W2: On a lack of availability of high-quality large-scale EEG-Text data**
>
> This is indeed currently an important topic. However, we would like to note that large amounts of this type of data exists in hospitals globally. Until now, there was little incentive to undertake the significant effort to share this data. We hope that promising research such as presented here may spur further data releases as method development progresses.
>
> > **W3: On sensitivity to text segmentation and the type of text input **
>
> It is indeed accurate that performance is strongly dependent on the text input, as this guides the pretraining process. We nevertheless included this broad set of results as we believed them to be informative for the field considering that this work is the first of its kind. Yet, when all text clusters are included, we observe highly stable performance across seeds, especially for the ELM-MIL models, which includes zero-shot performance across a wide range of temperature values (0.03 through 1.5, as may be seen in the extended Figure 3, Page 8).
>
> > **W4:  Fully supervised methods**
>
> We thank the reviewer for this suggestion and have added these to our TUAB, TUSZ, and TUEV evaluations, for which we use the same EEG encoder backbone as all other analyses. In most cases, we find our ELM-MIL models to considerably improve over a fully-supervised approach.
>
> > **W5: Other EEG-only self-supervised methods**
>
> We appreciate this concern and would like to briefly elaborate on our decisions. As we wanted to specifically compare pretraining strategies, we aimed to fix as many other factors as possible. This complicates comparisons against pretrained models (such as LaBraM) and methodologies relying on specific architectures. In order to avoid ambiguity whether performance differences were due to preprocessing, pretraining data, or encoder architecture, we selected two widely used methods as well as four EEG-specific methods that could be adapted for our setting to enable fair comparisons.
>
> > **Q1: The paper lacks a deeper discussion of the clinical implications (...)**
>
> We thank the reviewer for this important question regarding clinical implications. Our additional evaluations on the TUSZ and TUEV datasets demonstrate that the learned representations are effective at both recording-level classification and fine-grained event detection (5-second segments), suggesting future potential clinical utility for both diagnostic support and automated monitoring. While these results are promising, we acknowledge that further work is needed to make these representations more interpretable and clinically actionable. We have added a section titled 'Clinical implications (Section E, Page 24).
>
> > **Q2**:  On temporal dependencies and missing EEG segments.
>
> We hope the reviewer finds the additional TUSZ and TUEV evaluations informative. Indeed, probing the representations seems to miss segments compared to EEG-only models in terms of artefacts and eye movements, whereas clinical events were better classified using ELMs. Nevertheless, long-range temporal dependencies would constitute an interesting avenue for follow-up work, as this could be more explicitly modelled in our approach.
>
> We will address the final points in an additional comment.

---

> ### Author Response · Authors · 2024-11-19
> **Reply to Reviewer wPm6 (2 out of 2)**
>
> > **Q3: Regarding the normalization by number of crops in MIL loss. Is this appropriately addressing the varying amount of information in different-length crops and text segments?**
>
> The normalization was introduced primarily so as to balance the loss to depend on all subjects in a batch, as without it, subjects with short recordings or reports would contribute significantly more to the loss. Whereas recording length differs between subjects, the EEG crops are all equal length and we assume similar levels of information content, although MIL models can align some crops more than others. Regarding text segments, these indeed do differ in length. However, it is not necessarily the case that longer text segments have more information, as for example a 2-sentence segment can carry a lot of information by denoting the primary pathology.
>
> > **Q4:  I also have reservations over the procedure of selecting optimal value for temperature parameter. Why a separate held-out validation data is not used?**
>
> We thank the reviewer for allowing us to clarify. We did use a subsection of the TUAB training set as a validation set for temperature selection (Section B and B.4).  This data was afterwards included in the pretraining set to maximize the scarce amount of training data. The temperature ablations in Figure 3 rather serve to evaluate the stability of the models and our results. More elaborate hyperparameter selection could have been performed, but this would have also risked overfitting to the TUEG domain. We now also state in the main text (line 193) that we selected the temperature parameter using a holdout set and refer to section B.4.

---

### Author Response · Authors · 2024-11-19
**Global response to all reviewers**

We sincerely thank all reviewers for providing valuable feedback on our manuscript. We were delighted to read that reviewers appreciate our novel integration of sub-unit alignment and MIL to enable multimodal language modeling with EEG data, the improved pathology detection, and enabling zero-shot pathology classification and retrieval using brain activity signals for the first time.

&nbsp;

We are pleased to provide considerable extensions to the manuscript. We highlight the major additions here:

**1. External validation using the NMT Dataset**

**2. Additional evaluation: Seizure detection using TUSZ corpus**

**3. Additional evaluation: 6-class event detection using TUEV corpus**

**4. Improved model understanding through single-class performance visualizations**

**5. Addition of supervised baselines**

**6. Additional temperature ablations**

For your convenience, in addition to the revision we have collected the new tables and figures in [this anonymous document [link]](https://docs.google.com/document/d/e/2PACX-1vSsGqVFkvY1PEKzMxvVhHX-UiJ_r_1zS1dbF0RBf_Xlgj6D7QVwkncYkj2MXYR98sNT6bVA9Po9kpBl/pub).

&nbsp;

**1. External validation using the NMT Dataset - Section A.1 (Page 15)**

We use the NMT Scalp EEG Dataset from Pakistan as the second largest EEG corpus with abnormality labels to evaluate models out-of-distribution. This constitutes a challenging evaluation as the dataset is highly imbalanced and differs considerably from TUEG in terms of demographics and EEG recording setup. We find that our ELM-MIL approach still performs best in pathology detection across most settings.

&nbsp;

**2. Additional evaluation: Seizure detection using TUSZ - Section A.2 (Page 15)**

We evaluate models on the more challenging task of seizure detection based on a single, short EEG crop by pretraining all methods on 5-second crops. We observe that ELM-MIL exhibits performance improvements over EEG-only pretraining also in this considerably different evaluation.

&nbsp;

**3. Additional evaluation: 6-class event detection using TUEV - Section A.2 (Page 15+)**

TUEV poses a classification problem with three types of pathology events as well as eye movements, artefacts, and background activity, based on single 5-second EEG crops. We observe good performance of ELM-MIL on this task too.

&nbsp;

**4. Single-class performance plots - Figure 4 (Page 17)**

By investigating model performance per TUEV class, we find that ELM-MIL outperforms EEG-only pretraining for the important clinical events (spike and sharp waves as well as epileptiform discharges). However, the model underperforms for eye movement and artifact detection. Although such technical problems are described in some medical reports, this was segmented out in the current study. Future work could investigate whether including such text during pretraining could alleviate this shortcoming.

&nbsp;

**5. Addition of supervised baselines - Table 2 (Page 8) and Section D (Page 22)**

For the supervised baseline we use the same EEG encoder backbone as all other analyses, with an additional MLP classification head. We observe a similar pattern for fully supervised learning as reported in the literature. Namely, with few annotations, SSL can significantly improve upon supervised learning, with shrinking performance differences as more annotations come available. In comparison, our ELMs perform well even in the 100% label setting.

&nbsp;

**6. Additional temperature ablations - Figure 3 (Page 10)**

We provide additional evaluations at different temperature values, which now more clearly highlight that our MIL extension provides increased stability, especially for zero-shot classification.

---

### Meta-Review · Area_Chair_3PbM · 2024-12-23

**Metareview:**

This paper introduces EEG-Language Models (ELMs), a multimodal approach integrating EEG data with clinical text reports for pathology detection. The method employs time-series cropping, text segmentation, and a multiple-instance learning (MIL) variant of contrastive learning to address the alignment challenges between EEG signals and textual descriptions. Experimental results show improvements over EEG-only pretraining methods, with the ELM-MIL variant achieving the best performance in zero-shot and fine-tuning evaluations. The approach highlights the potential of multimodal pretraining in medical applications, offering richer representations.

Strength: The paper applies multimodal learning to EEG and clinical text reports to develop EEG-Language Models (ELMs), which is a quite interesting multimodal direction in this new domain of brain activity signals. It addresses the challenges of heterogeneity and length discrepancies between EEG and text data through sub-unit alignment and MIL. The results of ELMs demonstrate benefits over EEG-only pretraining on pathology detection tasks.

Weakness: The reviewers raise the concerns about the limited evaluation on a single dataset, comparison with supervised baselines, limited dataset scale, and ablation study on key parameters. Most importantly, the potential bias risks from the reliance on the pretrained language model and the alignment instability may stand out as the key concerns. Specifically, clinical reports may contain irrelevant information or fail to explicitly reference critical EEG features, leading to unstable multimodal alignment via contrastive learning. These issues may reduce the model's accuracy and reliability, limiting its effectiveness and practical utility in real-world clinical scenarios.

**Additional Comments On Reviewer Discussion:**

During the rebuttal, the author provided a very detailed response to all reviewers’ comments including adding additional evaluation tasks, supervised baselines and ablation studies, which are highly appreciated and effectively address some concerns as mentioned in the weakness. Most reviewers have responded to the rebuttal and one reviewer has raised the score to reflect the paper improvement from the author’s response and manuscript revision. Although the paper has been improved a lot by adding these new results, after reading the paper, reviewers’ comments and rebuttals, I think the fundamental question about the alignment of EEG and reports still exists as pointed out by multiple reviewers.

---

### Decision · Program_Chairs · 2025-01-22

Reject